# Highly Sensitive Gas Pressure Sensing with Temperature Monitoring Using a Slightly Tapered Fiber with an Inner Micro-Cavity and a Micro-Channel

**DOI:** 10.3390/s24216844

**Published:** 2024-10-24

**Authors:** Changwei Sun, Fen Yu, Huifang Chen, Dongning Wang, Ben Xu

**Affiliations:** 1School of Electronic and Electrical Engineering, Bengbu University, Bengbu 233030, China; 2College of Optical and Electronic Technology, China Jiliang University, Hangzhou 310018, China; 18767120234@163.com (F.Y.); chenhf@cjlu.edu.cn (H.C.); 3College of Unban Transportation and Logistics, Shenzhen Technology University, Shenzhen 518118, China; wangdongning@sztu.edu.cn

**Keywords:** optical fiber sensor, Mach–Zehnder interferometer, tapered fiber, gas pressure, temperature

## Abstract

A highly sensitive optical fiber gas pressure sensor with temperature monitoring is proposed and demonstrated. It is based on a slightly tapered fiber with an inner micro-cavity forming an in-fiber Mach–Zehnder interferometer (MZI), and a micro-channel is drilled into the lateral wall of the in-fiber micro-cavity using a femtosecond laser to allow gas to flow in. Due to the dependence of the refractive index (RI) of air inside the micro-cavity on its gas pressure and the high RI sensitivity of the MZI, the device is extremely sensitive to gas pressure. To prevent fiber breakage, the MZI is housed in a silicate capillary tube with an air inlet. Multiple modes are excited by slightly tapering the inner micro-cavity, and the resonance dips in the sensor’s transmission spectrum feature different linear gas pressure and temperature responses, so a sensitivity matrix algorithm can be used to achieve simultaneous demodulation of two parameters, thus resolving the temperature crosstalk. As expected, the experimental results demonstrated the reliability of the matrix algorithm, with pressure sensitivity reaching up to ~−12.967 nm/MPa and temperature sensitivity of ~89 pm/°C. The features of robust mechanical strength and high air pressure sensitivity with temperature monitoring imply that the proposed sensor has good practical and application prospects.

## 1. Introduction

Optical fiber gas pressure sensors have attracted much attention owing to their advantages such as compact constructure, resistance to electromagnetic interference, and high sensitivity, and they have been widely used in a variety of applications. To construct an optical fiber gas pressure sensor, various schemes, such as employing fiber Bragg grating (FBG) [1], a Fabry–Perot interferometer (FPI) [2,3,4,5], a Sagnac interferometer [6,7], or a Mach–Zehnder interferometer (MZI) [8,9,10], etc., can be utilized. According to their operation principle, generally, they can be divided into two types: One type is based on the pressure effect of gas, which causes device deformation or accompanied changes in the refractive index (RI), and the other type is based on the dependence of the gas RI on its pressure. For the first type, if gas pressure is applied directly to the optical fiber, the fiber’s deformation or RI variation is relatively minor due to its larger Young’s modulus and tiny optical-elastic coefficient, inevitably resulting in lower gas pressure sensitivity, typically a few pm/MPa [11]. To improve sensitivity, a range of membrane-based structures are proposed [5,12,13,14,15], as thin membranes are more susceptible to deformation under gas pressure. For example, at the end of the fiber, various thin film-based FPIs are constructed, whose cavity lengths are regulated by gas pressure, resulting in high pressure sensitivities. For instance, an FPI based on a silica film with a thickness of several micrometers achieved a sensitivity of 315 pm/MPa [14], a cured UV glue with good elasticity achieved a sensitivity of −40.94 pm/kPa [12], and a graphene film with a thickness of only a few nanometers achieved a sensitivity of up to 39.4 nm/kPa [16]. The thinner the membrane, the higher its sensitivity. However, a thin membrane makes the device more fragile, only appropriate for low-pressure measurements, and severely limits the measurement range. The second type works on the gas RI variation caused by its pressure. Essentially, it belongs to the category of RI sensors. Generally, an optical fiber refractometer exhibits a high RI sensitivity for an analyte whose RI is close to that of the optical fiber, while exhibiting an extremely low sensitivity for an analyte with a low RI, such as air gas [17,18,19]. In addition, the change in the gas RI caused by gas pressure is very tiny. Therefore, only sensors with sufficient sensitivity at the RI of air gas are capable of sensing gas pressure, such as open cavity-based FPIs with an RI sensitivity of ~1550 nm/RIU, which exhibit a gas pressure sensitivity of ~4 nm/MPa [2,3,4]. Compared to the first type, the second type of pressure sensor has a more robust structure and a larger measuring range of gas pressure, and there is still significant room for improvement in sensitivity. Another useful way to increase measurement sensitivity is spectral post-processing with the Vernier effect, which was proposed in recent years [7,20,21,22]. For example, two FPIs with similar cavity length were connected in series, with one serving as a pressure sensor and the other as a reference. Using data post-processing based on the Vernier effect, ultra-high sensitivity of up to 283.32 nm/kPa was achieved [7]. However, because the measurement error is amplified when the sensitivity is amplified, the measurement accuracy is not effectively improved in essence [23]. To ensure measurement accuracy, in addition to improving the sensitivity of the original direct measurement, it is critical to minimize the influence of ambient factors such as temperature cross-sensitivity. For example, a pressure insensitive FBG is connected in series to a gas pressure sensor to monitor the surrounding temperature and compensate for the impact of temperature on pressure measurement [3]. Multi-FPI is another commonly used structure for gas pressure measuring with temperature compensation. For instance, a segment of single-mode fiber is spliced to a capillary tube embedded with a microsphere to construct a multi-FPI structure, where the FPI with an air-cavity is sensitive to gas pressure sensor while the FPI composed of the microsphere is sensitive to temperature [24]. To compensate for the temperature effect, the multiple-dip tracing technique and the Fourier band pass filtering (FBPF) method are used for simultaneous measurement of gas pressure and temperature [4]. To some extent, combining discrete components complicates the sensing system and data processing, while using a single sensor to achieve simultaneous measurement of multiple parameters can make the sensor more compact, which is also an aim pursued by researchers [25].

In this paper, a simultaneous gas pressure and temperature measurement sensor based on an in-fiber MZI is designed and experimentally validated. The device comprises a slightly tapered fiber with an inner micro-cavity, and a micro-channel created by a femtosecond laser in the side wall of the micro-cavity, allowing gas to flow in. The RI of gas varies with the change in gas pressure, and the device is highly sensitive to gas pressure. On the other hand, the device is also sensitive to surrounding temperature due to the dependance of the effective RI of the micro-cavity wall on the temperature. Due to the presence of multiple modes excited in this MZI, each mode exhibits different linear responses to gas pressure and temperature. Therefore, a straightforward sensitivity matrix approach can be employed to simultaneously demodulate these two parameters.

## 2. Sensing Principle

Figure 1 shows the schematic of the device. A lightly tapered fiber is sealed in a silicate capillary tube with an air inlet with glue, and there is an inner air-cavity with a side-opened micro-channel at the waist of the tapered fiber, allowing gas to easily flow into the air-cavity.

Light confined and propagated in the core of the lead-in fiber is split into two beams when it arrives at the air-cavity because of the down taper and curved air-glass interface; one is propagated forward in the air-cavity, while the other leaked into the clad and propagated in the lateral wall of the air-cavity. Then, they are combined through the air-cavity and propagated in the lead-out fiber. Thus, this structure described in Figure 1 can be considered as a typical in-fiber MZI, and it can be characterized by the two-beam optical interference equation as [26]
(1)$$I\;=\;I_{air}\;+\;I_{clad}+2\sqrt{I_{air}I_{clad}}cos\left(∆\varphi \right),$$
where *I* is the transmitted light intensity of the MZI, $I_{air}$ and $I_{clad}$ indicate the intensities of light propagating in the air-cavity and the lateral wall, respectively; ∆φ is the phase difference between these two beams of light, and $∆\varphi \;=\;2\pi L∆n_{eff}/\lambda$, where *L* represents the effective interference length which is close to the longitudinal length of the air-cavity, $∆n_{eff}\left(=n_{eff}^{clad}\;-\;n_{eff}^{air}\right)$ is the difference between the effective refractive indices of air and fiber clad (since the lateral wall of the air-cavity is made of fiber clad), *λ* is the wavelength of the incident light in vacuum. When ∆φ satisfies an odd multiple of π, the destructive interference is then triggered, and the transmitted intensity achieves a minimum value with a resonance dip in the transmission spectrum, and the central wavelength of the dip can be obtained as
(2)$$\lambda_{m}\;=\;\frac{2L∆n_{eff}}{2m\;+\;1}.\;(m\;=\;0,\;1,\;2,\;3\ldots \ldots )$$

Further, the refractive index (RI) of gas depends on its pressure and temperature. For instance, at room temperature, the RI of air gas can be expressed as [3]:(3)$$n\;=\;1\;+\;\frac{2.8793\;\times \;10^{-3}\;\times \;P}{1\;+\;0.003671\;\times \;T},$$
where *P* and *T* indicate the gas pressure and temperature in units of Pascal (Pa) and Celsius degree (°C), respectively. For a constant of temperature such as 20 °C, the variation of gas RI with pressure, dn/dP is ~2.68 × 10^−3^/MPa. According to Equations (2) and (3), the change in gas pressure induces a change in the RI of the gas, which in turn causes a change in $∆n_{eff}$ and ultimately a shift in the resonance dip, and the following equation represents the relationship between them:(4)$$\frac{d\lambda_{m}}{dP}\;=\;\frac{d\lambda_{m}}{dn}\cdot \frac{dn}{dP}\;=\;-\frac{\lambda_{m}}{n_{eff}^{clad}\;-\;n_{eff}^{air\;}}\cdot \frac{dn}{dP}.$$

Thus, theoretical gas RI sensitivity, $d\lambda_{m}/dn$, of ~−3491 nm/RIU (refractive index unit) and pressure sensitivity, $d\lambda_{m}/dP$, of −9.36 nm/MPa can be expected based on the resonance dip wavelength of 1550 nm and the clad RI of 1.444.

## 3. Sensor Fabrication

To fabricate the proposed device described above, several steps are involved. Figure 2 illustrates the fabrication process. Step 1: A small pit is drilled in the end face of a single-mode fiber (SMF, YOFC) S1 cleaved flatly. To achieve that, a femtosecond laser (fs-laser) with a pulse energy of 2 μJ, a pulse duration of 35 fs, and a repetition rate of 5 kHz is focused on the core area of the end face by using an objection lens with an NA of 0.45, as shown in Figure 2a, resulting in a small pit with a diameter of ~4 μm and depth of ~5 μm. Step 2: The SMF S1 with a small pit in the end face is fusion spliced to another SMF S2 with a flat end face, as shown in Figure 2b. Due to the expansion of the air gas in the pit under arc heating, a micro-cavity is formed in the fiber by adopting appropriate fusion splicing parameters such as arc discharge current and time, and thus a well-known in-fiber air-cavity-based Fabry–Perot interferometer is obtained [4]. Step 3: The fiber with an inner micro-cavity is slightly tapered by stretching it at both ends in opposite directions while it is heated by a flame, as shown in Figure 2c. By setting appropriate stretching speed and distance, the cavity length can be precisely controlled, which directly determines the excitation of the cladding modes and the interference length of the device, and the free spectrum range (FSR) in its transmission spectrum. To prevent the collapsing of the air-cavity, weaker flame intensity and a shorter heating time are necessary. It should be noted that during the process, the transmission spectrum of the device is continuously monitored until the desired interference fringes are observed, then tapering is stopped. Thus, a fiber-in micro-cavity-based MZI is obtained. Step 4: A micro-channel is drilled in the lateral wall of the inner micro-cavity by using a femtosecond laser, allowing the outside air to flow in, as shown in Figure 2d. During the process, there is no need to precisely control the parameters of femtosecond laser micromachining. Step 5: The fabricated fiber-in MZI is housed in a silicate capillary tube with an adhesive of AB glue to improve the mechanical strength and practicality of the sensor, as shown in Figure 2e. In our experiments, the capillary tube features an inner diameter of 300 µm and an outer diameter of 900 µm, and an air inlet in the lateral wall. Figure 2f shows a microscopic image of the fiber-in MZI encapsulated in the capillary tube. Due to the tapering process, the inner air-cavity is elliptical in shape, and a micro-channel is clearly visible in its lateral wall. Owing to the precise digital control throughout the preparation process, including arc discharge current and time, stretching speed and distance for tapering fiber, etc., the preparation of in-fiber micro-cavity is reproducible.

Figure 3 depicts the reflection and transmission spectra measured during the sensor preparation process, namely before and after tapering and drilling the micro-channel. Figure 3a depicts the reflection and transmission of the device generated via fusion splicing, corresponding to Figure 2b, prior to tapering. The transmission spectrum is virtually a straight line, with no noticeable interference fringes, whereas the reflection spectrum has obvious periodic fringes and a uniform extinction ratio. This result is perfectly consistent with the typical properties of fiber-in FP interferometers [4]. Figure 3b depicts the device’s reflection and transmission spectra prior to and following drilling of the micro-channel in the lateral wall after it has been tapered. In contrast, after tapering, the transmission spectra show obvious periodic fringes with an extinction ratio of more than 10 dB, whereas the reflection spectra show no obvious interference fringes, which can be explained by Mach–Zehnder interference caused by deformation of the inner micro-cavity after the fiber is tapered. Furthermore, we can see that the device’s reflection and transmission spectra differ slightly before and after drilling the micro-channel, demonstrating that the micro-channel has little effect on the device’s reflection and transmission spectra due to its small size.

Then, the strain response of the fiber-in MZI was tested. The fiber-in MZI was clamped with two translation stages with a separation distance of ~27 cm. Strain was applied to it by moving one stage to elongate the MZI along its longitudinal axis, while the other one was kept fixed. Figure 4a presents the transmission of the device under different strains. As can be seen, its transmission spectrum exhibits a blue shift with the increase in strain. Furthermore, Figure 4b gives the detailed wavelength shifts of the tracking resonance dip for different strains relative to the initial position at 1555.356 nm corresponding to zero strain, and it is found that there is a good linear relation between them by employing a least squares fitting with R^2^ of ~0.9982, and the slope of the fitting function indicates a strain sensitivity of ~−5.04 pm/µε within the range of 0–756 µε, which is close to the previously reported sensitivity of a similar structure without a micro-channel in the lateral wall [26].

## 4. Experiments and Analysis

Figure 5 shows the schematic diagram of the proposed sensor with the capillary tube enveloped for gas pressure sensing. The sensor was placed into a gas chamber which was connected to an air pump, and the gas pressure in the chamber could be precisely controlled in steps of 0.04 MPa with the help of a pressure meter. Meanwhile, a homemade broadband light source (BBS) covering C + L band and an optical spectrum analyzer (OSA, YOKOGAWA, Japan, AQ6370D) with a wavelength resolution bandwidth of 0.02 nm and a sampling interval of 0.01 nm were connected to the both ends of the sensor, respectively, then the transmission spectrum of the sensor could be measured in real-time. In our experiments, the differential gas pressure in chamber compared to outside varied from 0 to 0.8 MPa, with a step interval of 0.04 MPa. The insert in Figure 5 gives a micrograph of the sensing head. Due to the fiber being pulled slightly under flame, the waist diameter of ~84 µm is thinner than that of a standard SMF; the length of the long and short axis of the air-cavity are ~188 µm and ~56 µm, respectively. The diameter of the micro-channel is measured to be ~13 µm.

Figure 6a depicts the response of the sensor under different differential gas pressures. It is clearly shown that the transmission spectrum experiences a blue shift of ~9 nm when the differential gas pressure in the chamber increases from 0 to 0.8 MPa within the wavelength range of 1520 nm to 1580 nm. Furthermore, by tracking the center wavelength of the resonance dip at different pressures and applying the least squares method for linear fitting, as shown in Figure 6b, it is found that there is a good linear relationship between the resonance dip wavelength shift relative to the initial position at 1556.875 nm corresponding to zero differential gas pressure (i.e., the gas pressure inside the chamber is equal to the atmospheric pressure outside the chamber) and the pressure with R^2^ of 0.9990, and the slope of the linear curve achieves up to −10.77 nm/MPa in characterizing the sensitivity of the sensor, which is much higher than that of many other fiber sensors based on FPI or MZI [2,3,4,11,14]. As analyzed in Equation (4) above, the RI of gas in the air-cavity increases with increasing gas pressure, leading to a spectrum shift with a theoretical sensitivity of −9.36 nm/MPa, which is very close to the measured value. The margin between theoretical and measured values may be attributable to the tiny deformation of the air-cavity. As gas pressure in the air-cavity rises, the air-cavity inevitably expands axially, applying axial strain on the fiber-in MZI. According to the experimental results shown in Figure 4, the fiber-in MZI exhibits a small blueshift strain sensitivity, which increases the blue shift of the device’s transmission spectrum when gas pressure increases, explaining the difference between the measured and theoretical sensitivities.

Then, the temperature response of the fiber-in MZI was tested by placing it in an electric furnace. The furnace works by means of a thermoelectric cooler (TEC) with functions of cooling and heating, and a temperature resolution of 0.1 °C. The temperature varied from 15 °C to 65 °C in intervals of 5 °C, and the temperature was maintained for five minutes at each step until the transmission spectrum of the device reached stability. Figure 7a shows the transmission spectra of the fiber-in MZI under different temperatures. It is clearly shown that the transmission spectrum exhibits a red shift with the increase in ambient temperature, tracing a resonant dip near 1556 nm, and the total shift reached ~1.8 nm for a temperature variation of 50 °C. Furthermore, Figure 7b presents the specific dip wavelength shift relative to the initial position at 1556.416 nm, corresponding to 15 °C versus temperature, and it is found there is a good linear relationship between them with a linear correlation coefficient of 0.9965. The slope of the fitting function indicates the device’s temperature sensitivity to be ~36.25 pm/°C, which is higher than that of most gas pressure sensors based on FPIs [2,3,4]. This means there is a certain impact of environmental temperature on the measurement of gas pressure.

To overcome the temperature cross-sensitivity, monitoring environmental temperature is particularly necessary for accurately measuring gas pressure. Our previous research found that several orders of cladding mode can be excited by changing the structural parameters of a slightly tapered optical fiber with an inner air-cavity, resulting in a number of resonance dips in the transmission spectrum of the device, which allows for the sensing of simultaneous multiple parameters by monitoring the variation of selected different resonance dips [17]. Then, another fiber-in MZI was fabricated for achieving simultaneous gas pressure and temperature measurement. Figure 8 presents the device’s reflection and transmission spectra prior to and following drilling of a micro-channel in the lateral wall after it has been tapered. Like the previous in-fiber MZI, the micro-channel has little effect on the transmission or reflection spectra trends seen in the device; the transmission spectra show obvious fringes, whereas the reflection spectra show no obvious interference fringes. By contrast, the resonance dips feature significantly different extinction ratios, indicating that different orders of cladding modes are indeed excited [17]. The inset in Figure 8 shows a micrograph of the second fabricated fiber-in MZI. The device features a taper waist diameter of ~91 µm, and the long and short axis lengths of the air-cavity are ~156 µm and ~62 µm, respectively. The diameter of the micro-channel is ~11 µm. Then, the second fiber-in MZI is housed in another silicate capillary tube to form a new sensor.

Figure 9 presents the gas pressure response of the second sensor. With the increase in gas pressure, the transmission spectrum exhibits an obvious blue shift, and there is a large difference between the wavelength shifts for resonance dip A at ~1510 nm and dip B at ~1550 nm, with the former being ~8 nm and the latter ~13 nm, while the differential gas pressure varies from 0 to 1.0 MPa, as shown in Figure 9a. Figure 9b gives the specific resonance dip wavelength shifts of dip A and dip B relative to their initial positions at 1509.426 nm and 1552.994 nm, corresponding to zero differential gas pressure for different differential gas pressures. The least squares fitting indicates both of the resonance dips linearly shift with increasing differential gas pressure, and they exhibit different sensitivities according to the different slopes of the fitting curves of −8.315 nm/MPa and −12.967 nm/MPa, respectively.

In addition, the temperature response of the second sensor was investigated. As ambient temperature rises from 20 °C to 80 °C, the device’s transmission spectrum exhibits a red shift, as illustrated in Figure 10a. Similarly, least squares fitting is used to explore the relationship between them. Figure 10b shows the fitting results for resonance dip wavelength shifts relative to their initial positions at 1509.452 nm and 1547.715 nm, corresponding to 15 °C versus ambient temperature. Clearly, the two resonance dips have different temperature sensitivities of 89.0 pm/°C and 71.6 pm/°C.

As observed in Figure 9 and Figure 10, within the measured ranges, the resonance dips A and B exhibit different linear responses to gas pressure and temperature, and they do not interfere with each other, which exactly satisfies the conditions for constructing a matrix of order two for simultaneously measuring pressure and temperature. The wavelength shifts of dip A and B corresponding to the variation of gas pressure and temperature, respectively, can be written by using a matrix as:(5)$$\left[\begin{array}{c} ∆\lambda_{A}\\ ∆\lambda_{B} \end{array}\right]\;=\;\left[\begin{array}{cc} k_{A,P} & k_{A,T}\\ k_{B,P} & k_{B,T} \end{array}\right]\left[\begin{array}{c} \Delta P\\ \Delta T \end{array}\right],$$
where $∆\lambda_{A}$ and $∆\lambda_{B}$ are the wavelength shifts, $k_{A,P}$ and $k_{B,P}$ represent the gas pressure sensitivities, and $k_{A,T}$ and $k_{B,T}$ stand for the temperature sensitivities of dip A and B, respectively. ΔP and ΔT denote the variation of gas pressure and temperature applied onto the device, respectively. Based on the measured sensitivity coefficients, a demodulation matrix can be established to obtain gas pressure and temperature values:(6)$$\left[\begin{array}{c} \Delta P\\ \Delta T \end{array}\right]\;=\;{\left[\begin{array}{cc} k_{A,P} & k_{A,T}\\ k_{B,P} & k_{B,T} \end{array}\right]}^{-1}\left[\begin{array}{c} ∆\lambda_{A}\\ ∆\lambda_{B} \end{array}\right]\;=\;{\left[\begin{array}{cc} -8.315 & 0.0890\\ -12.967 & 0.0716 \end{array}\right]}^{-1}\left[\begin{array}{c} ∆\lambda_{A}\\ ∆\lambda_{B} \end{array}\right]$$

To prove the validity of Equation (6), a simultaneous gas pressure and temperature measurement performance experiment was implemented for the sensor. Two sets of experiments were carried out: applying the differential gas pressure in the range from 0 to 1.0 MPa at a temperature of 26 °C, and varying the temperature under a specific differential gas pressure of 0.8 MPa. The results obtained are displayed in Figure 11. It is found that the calculated values according to the matrix presented in Equation (6) are very close to the values applied with maximum errors of ±28 kPa, ±1.2 °C for gas pressure and temperature, respectively, which indicates that the matrix method is valid within the range of 0–1.0 MPa, and that the resolution of the sensor is approximately 28 kPa and 1.2 °C for gas pressure and temperature sensing, respectively. In our experiments, the precision values of the pressure meter and the electric furnace were 40 kPa and 0.1 °C, respectively. Due to the similar thermal-optic coefficients of cladding modes, the temperature sensitivity between the two resonance dips is small; as such, the value of $\left(k_{A,P}∆k_{B,T}\;-\;k_{B,P}∆k_{A,T}\right)$ is small, resulting in a relative larger temperature measurement error. Using the support vector regression method to generate a nonlinear function for the entire region using a statistical machine learning process may be helpful in reducing measurement errors [27].

Compared with other optical fiber gas pressure sensors with different structures or working principles, as shown in Table 1, our sensor has certain advantages in terms of performance such as sensitivity, temperature compensation, and robustness. For the sensors based on a fiber’s RI changes generated by pressure, because of the silica fiber’s high Young’s modulus and small elastic-optical coefficient, the sensitivities of such sensors are typically as low as tens of pm/MPa [1,11]. Unlike them, our sensor operates on the gas’s RI changes when modulated by gas pressure, and it is extremely sensitive to these RI changes; our sensor has a maximum gas pressure sensitivity of 12.967 nm/MPa, which is two orders of magnitude higher than the previous type, and more than twice as high as FPI-based gas pressure sensors with similar working principles [2,3,4]. Even compared to the MZI-based pressure sensors with the same working principle but different structures [8,9,10], such as in-fiber MZI based on large lateral offset fusion splicing [28], our sensor has better robustness and higher sensitivity. Compared with sensors based on especial MZI [29] or diaphragm deformation caused by pressure [12,16,30], our sensor has relatively lower pressure sensitivity, but can withstand higher pressure with wider measurable range and provide temperature compensation eliminating the problem of temperature cross-sensitivity, which is crucial for accurate pressure sensing. For the scheme of connecting an additional FBG for temperature compensation, our sensor is more compact [3]. The post-processing algorithm based on the Vernier effect is effective in considerably improving sensitivities, but the sensor’s actual detection limit is not improved or even decreased, implying that there is no substantial contribution to measurement accuracy [23].

Notably, in contrast to the previously investigated in-fiber MZI with an inner cavity [17], the sensor’s distinction resides in the additional micro-channel in the lateral wall of the micro-cavity. This micro-channel markedly enhances its RI sensitivity by two orders of magnitude, making it possible for gas pressure sensing. In the absence of the micro-channel, the effective RI of the fiber clad is only marginally influenced by the surrounding gas; however, with the micro-channel, air gas flows into the micro-cavity, and the change in gas RI is completely converted into the effective RI change of the in-fiber MZI’s sensing arm. Therefore, the gas RI sensitivity is greatly improved, which can also be explained by Equation (4).

## 5. Conclusions

A highly sensitive optical fiber gas pressure sensor with temperature monitoring is proposed and experimentally validated. It is made of a slightly tapered fiber with an inner micro-cavity forming an in-fiber MZI, as well as a micro-channel in the lateral wall of the micro-cavity that allows gas to flow in. The operation principle and fabrication process are described in detail. Because the gas RI is pressure dependent, and the in-fiber MZI has extremely high RI sensitivity, this sensor can detect gas pressure with great sensitivity. Due to the excitation of multiple modes caused by slightly tapering the inner micro-cavity, a sensitivity matrix algorithm can simultaneously demodulate pressure and temperature with maximum sensitivities of ~−12.967 nm/MPa and ~89 pm/°C, and maximum errors of 28 kPa and 1.2 °C within the range of 0–1.0 MPa and 15–80 °C, respectively. The features of robust structure and high gas pressure sensitivity with temperature compensation imply that the proposed sensor has good practical and application prospects in various areas such as environmental safety monitoring, aerospace, military, and medical diagnosis.

## Figures and Tables

**Figure 1 sensors-24-06844-f001:**
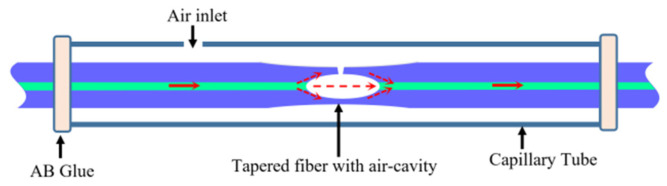
Schematic diagram of the gas pressure sensor.

**Figure 2 sensors-24-06844-f002:**
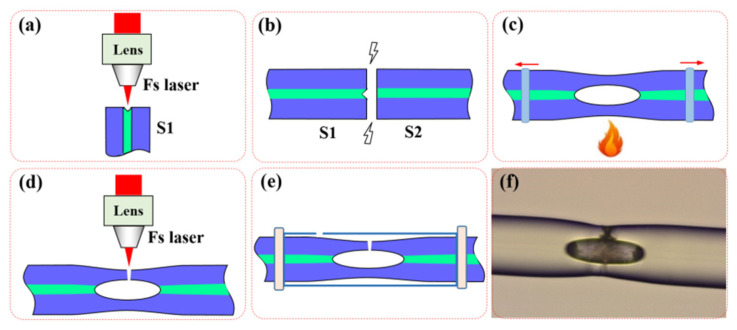
The fabrication process of the gas pressure sensor. (**a**) A small pit is etched on the end face of the SMF by using Fs-laser. (**b**) The etched fiber is spliced to another SMF with a flat end face. (**c**) The fiber with an inner micro-cavity is tapered by streching. (**d**) A micro-channel is drilled in the lateral wall of the micro-cavity by using Fs-laser. (**e**) The fiber-in MZI is housed in a silicate capillary tube. (**f**) Microscopic image of the fiber-in MZI encapsulated in the capillary tube.

**Figure 3 sensors-24-06844-f003:**
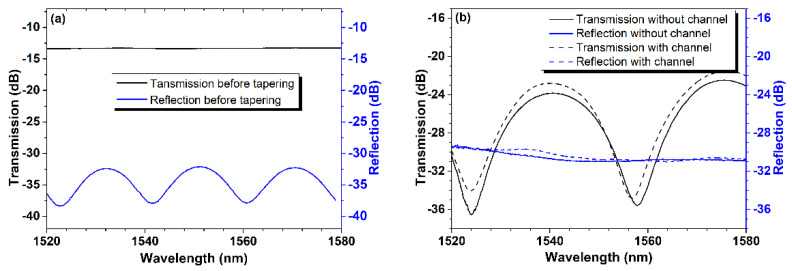
Reflectance and transmission spectra monitored during the device preparation process. (**a**) The reflection and transmission spectra of the non-tapered device after the preparation steps corresponding to Figure 2b. (**b**) Reflection and transmission spectra of the slightly tapered device before and after drilling the micro-channel.

**Figure 4 sensors-24-06844-f004:**
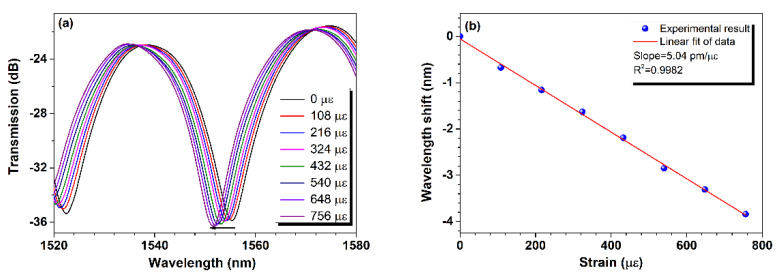
Strain response of the fiber-in MZI (**a**) Transmission spectra of the device under different strains and (**b**) dip wavelength shift versus strain.

**Figure 5 sensors-24-06844-f005:**
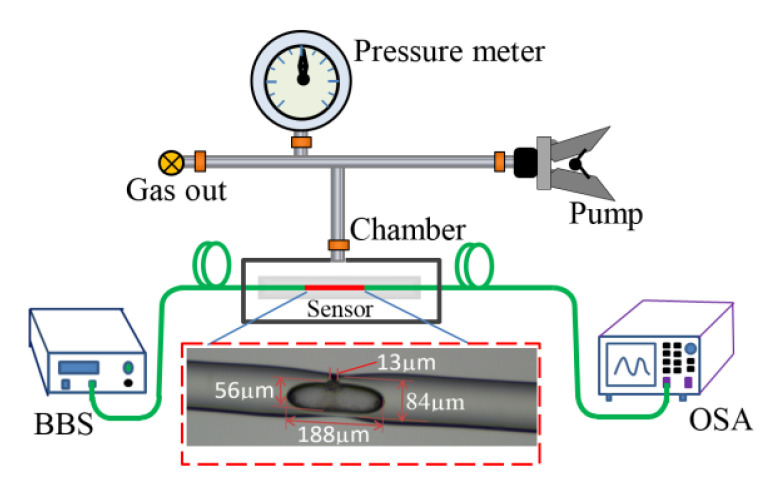
Experimental setup for investigating the response of the sensor to gas pressure.

**Figure 6 sensors-24-06844-f006:**
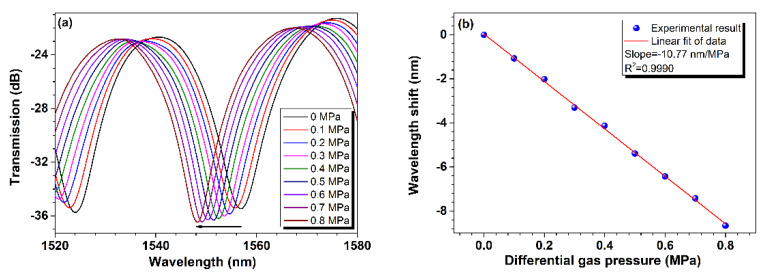
Pressure response of the sensor with capillary enveloped. (**a**) Transmission spectra of the sensor under different pressures; (**b**) dip wavelength shift versus gas pressure. Correlation between the resonance wavelength shift and the differential gas pressure.

**Figure 7 sensors-24-06844-f007:**
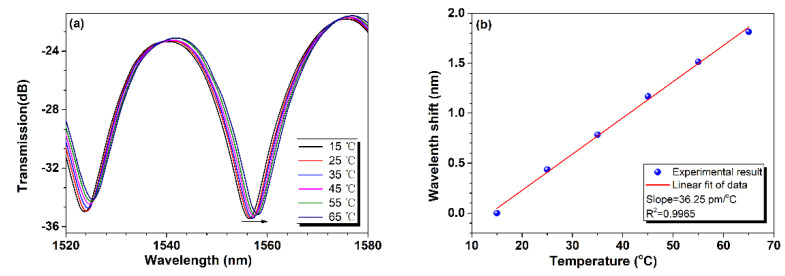
Temperature response of the fiber-in MZI. (**a**) Transmission spectra under different temperatures; (**b**) dip wavelength shift versus temperature.

**Figure 8 sensors-24-06844-f008:**
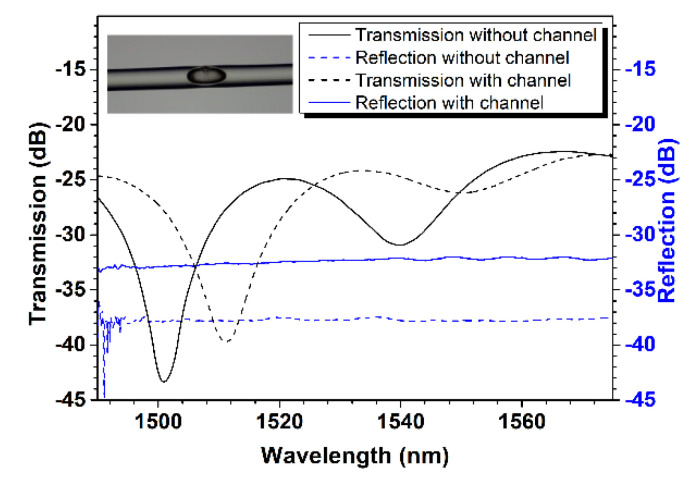
Reflection and transmission spectra of the second fiber-in MZI before and after drilling the micro-channel.

**Figure 9 sensors-24-06844-f009:**
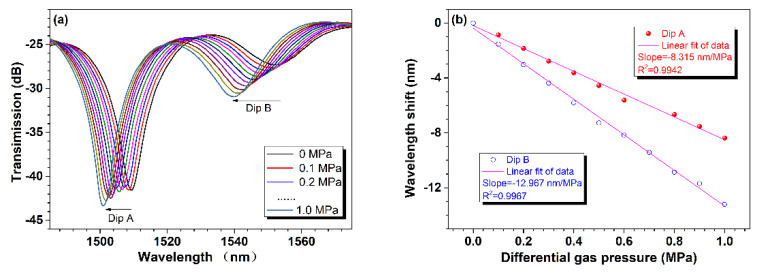
Gas pressure response of the second sensor. (**a**) Reflection spectral of the sensor at different gas pressures, and (**b**) wavelength shifts of dip A and B versus differential gas pressure.

**Figure 10 sensors-24-06844-f010:**
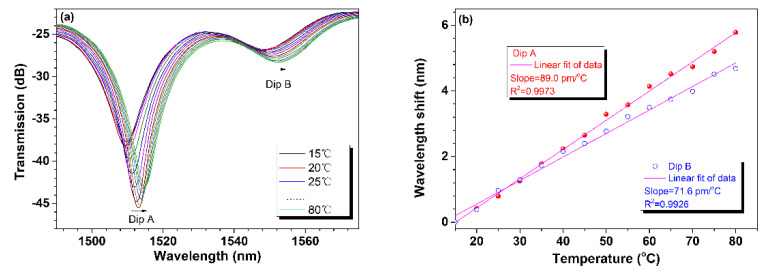
Temperature response of the second sensor. (**a**) Reflection spectral of the sensor at different temperatures, and (**b**) wavelength shifts of dip A and B versus temperature.

**Figure 11 sensors-24-06844-f011:**
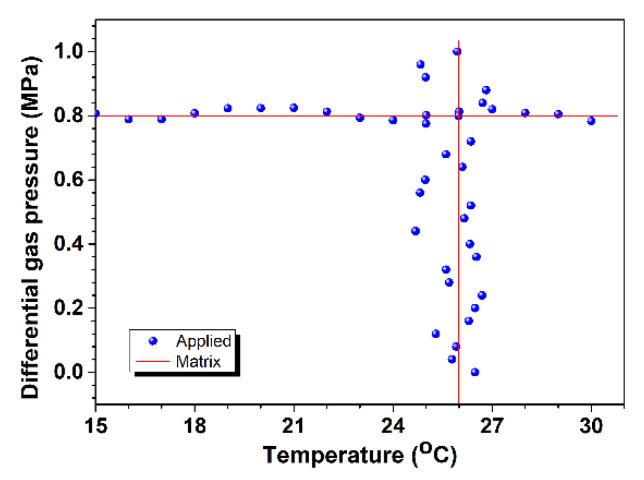
Results of simultaneous gas pressure and temperature.

**Table 1 sensors-24-06844-t001:** Comparison with other gas pressure sensors (SM: Simultaneous measurement).

Sensor Structure	Strategy	Gas Pressure Sensitivity	Temperature Sensitivity	Cross Sensitivity
FBG [1]	Deformation	79.7 pm/MPa	9.64 pm/°C	121 kPa/°C
FPI [11]	Deformation	~5.8 pm/MPa	~14 pm/°C	2.414 MPa/°C
FPI [2]	RI effect	4.147 nm/MPa	1.18 pm/°C	0.3 kPa/°C
FPI-FBG [3]	RI effect	4.14 nm/MPa	12 pm/°C	SM
Multi-FPI [4]	RI effect	4.028 nm/MPa	7.1 pm/°C	SM
Multimode interferometer [8]	RI effect	8.1 nm/MPa	12.3 pm/°C	1.5 kPa/°C
MZI [9]	RI effect	8.239 nm/MPa	45.6 pm/°C	5.5 kPa/°C
MZI [10]	RI effect	9.6 nm/MPa	43 pm/°C	4.4 kPa/°C
FPI [12]	Deformation	40.94 nm/MPa	213 pm/°C	5.2 kPa/°C
FPI with nanofilm [16]	Deformation	39.4 nm/kPa	--	--
MZI [29]	RI effect	2.39 nm/kPa	--	--
MZI [30]	RI effect	1.27 nm/kPa	Avage: 0.67 nm/kPa	0.528 kPa/°C
MZI (This work)	RI effect	12.967 nm/MPa	89 pm/°C	SM

## Data Availability

All data included in this study are available upon request by contact with the corresponding author.

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
