# Peer review of "Highly Sensitive Gas Pressure Sensing with Temperature Monitoring Using a Slightly Tapered Fiber with an Inner Micro-Cavity and a Micro-Channel"

_sensors, 2024, doi:10.3390/s24216844_

Round 1
Reviewer 1 Report
Comments and Suggestions for Authors
The research demonstrates a novel method for simultaneously measuring gas pressure and temperature using slightly tapered fiber sensor with an inner micro-cavity and a micro-channel. The technique emphasises the sensor's exceptional capability to reduce the interference between gas pressure and temperature through the use of multi-mode interference. The article can be accepted with a minor modification.
The following are recommendations for enhancement:
1. In the introduction, the application of pressure and temperature dual parameter sensing should be provided.
2. What is the time for slightly taper processing?
3. In line 192, does a variation in air pressure induce deformation of the micro-cavity, leading to spectral drift?
4. The resolution of the sensor should be provided.
5. The annotation in Figure 10 is inaccurate. The line should depict the applied temperature and pressure, and the scatter plot represents the result calculated through a matrix.
6. The comparison with other optical fiber gas pressure sensors should be conducted by utilizing tables or specific numerical parameters.
7. Illustrating the practical utility of this sensor would involve discussing its potential real-world applications and presenting case studies or examples of where it has been or may be effectively used.
Reviewer 2 Report
Comments and Suggestions for Authors
The manuscript proposes and fabricates an in-line slight-tapered air cavity Mach-Zehnder fiber interferometer sensor with microchannels. The sensor is applied to simultaneous sensing experiments of temperature and air pressure, demonstrating excellent multimodal sensing performance. The main innovation of the manuscript lies in the structural design of the sensing head. After addressing the following issues, the manuscript is recommended for publication in this journal:
- Regarding the air cavity in the sensing head, how are certain physical parameters (e.g., cavity length or thickness) and the shape of the cavity controlled, and how do they affect sensing performance? Is it possible to replicate the preparation?
- In Figure 5(b), the vertical axis represents the wavelength shift related to air pressure. Please provide the specific value of the initial reference wavelength (resonant dip) in the text, as explained in Figures 6(b) and 8(b).
- The authors mention multiple times (lines 199, 215, 293-312) that the sensor’s performance is better than that of other similar sensors. Please provide an intuitive table comparing the performance differences of these sensors in the manuscript.
Reviewer 3 Report
Comments and Suggestions for Authors
The work has overlaps with previously published works:
1. B. Xu, Y. Liu, D. Wang, D. Jia and C. Jiang, "Optical Fiber Fabry–Pérot Interferometer Based on an Air Cavity for Gas Pressure Sensing," in IEEE Photonics Journal, vol. 9, no. 2, pp. 1-9, April 2017, Art no. 7102309, doi: 10.1109/JPHOT.2017.2685939. keywords: {Cavity resonators;Sensitivity;Temperature sensors;Optical fiber sensors;Optical fibers;Sensors;interferometer;Fourier band pass filtering (FBPF);gas pressure.},
2. B. Xu, Y. M. Liu, D. N. Wang, and J. Q. Li, "Fiber Fabry–Pérot Interferometer for Measurement of Gas Pressure and Temperature," J. Lightwave Technol. 34, 4920-4925 (2016).
https://opg.optica.org/jlt/abstract.cfm?URI=jlt-34-21-4920
Therefore, I can not recommend this manuscript for publication.
Reviewer 4 Report
Comments and Suggestions for Authors
1. The novelty in the study is not clearly articulated and needs further elaboration. 2. The literature review lacks specificity and requires more detailed and focused analysis. 3. Figure 5 is not presented in a manner that facilitates easy comprehension and needs to be clarified. 4. The rationale behind the application of equations 5 and 6 in the results requires more thorough elucidation. 5. The conclusion should be enriched with additional numerical data to provide a more comprehensive summary of the results.
Round 2
Reviewer 3 Report
Comments and Suggestions for Authors
The manuscript provides design, fabrication steps, and experimental demonstration of air and temperature sensors. However, I do not find a significant contribution from this manuscript, as similar works have been published with comparable sensitivity. The work has practical value, but the technique is not innovative enough to be recommended for publication in this journal. The method is old, and no significant sensitivity enhancement has been achieved.
1. W. Talataisong, D. N. Wang, R. Chitaree, C. R. Liao, and C. Wang, "Fiber in-line Mach–Zehnder interferometer based on an inner air-cavity for high-pressure sensing," Opt. Lett. 40, 1220-1222 (2015)
2. T. Y. Hu, Y. Wang, C. R. Liao, and D. N. Wang, "Miniaturized fiber in-line Mach–Zehnder interferometer based on inner air cavity for high-temperature sensing," Opt. Lett. 37, 5082-5084 (2012)
3. Optical fiber Mach-Zehnder interferometer based on a pair of microholes. https://doi.org/10.1016/j.yofte.2022.103016.
4. Better sensitivity: Chun Mao, Bo Huang, Ying Wang, Yijian Huang, Longfei Zhang, Yu Shao, and Yiping Wang, "High-sensitivity gas pressure sensor based on hollow-core photonic bandgap fiber Mach-Zehnder interferometer," Opt. Express 26, 30108-30115 (2018).
5. Hourglass-shaped fiber-optic Mach-Zehnder interferometer for pressure sensing. https://doi.org/10.1016/j.yofte.2024.103746.
Author Response
Dear reviewer, thanks for your helpful opinions. According to your opinions, we have further revised the manuscript, please see the attached.
